# Does the Immunohistotype of Breast Cancer Influence Ovarian Reserve and Fertility Preservation Outcomes? A Single-Center Prospective Observational Study

**DOI:** 10.3390/cancers17213564

**Published:** 2025-11-03

**Authors:** Valentina Immediata, Sabrina Aprea, Emanuela Morenghi, Annamaria Baggiani, Cristina Specchia, Damiano Gentile, Corrado Tinterri, Paolo Emanuele Levi-Setti

**Affiliations:** 1Fertility Center, Humanitas Research Hospital, IRCCS, Via Manzoni 56, Rozzano, 20089 Milan, Italy; valentina.immediata@humanitas.it (V.I.); sabrina.aprea@humanitas.it (S.A.); annamaria.baggiani@humanitas.it (A.B.); cristina.specchia@humanitas.it (C.S.); paolo.levi_setti@hunimed.eu (P.E.L.-S.); 2Biostatistics Unit, Humanitas Research Hospital, IRCCS, Via Manzoni 56, Rozzano, 20089 Milan, Italy; emanuela.morenghi@humanitas.it; 3Breast Unit, Humanitas Research Hospital, IRCCS, Via Manzoni 56, Rozzano, 20089 Milan, Italy; corrado.tinterri@hunimed.eu; 4Department of Biomedical Sciences, Humanitas University, Via Rita Levi Montalcini 4, Pieve Emanuele, 20090 Milan, Italy

**Keywords:** breast cancer, fertility preservation, ovarian reserve, controlled ovarian stimulation, triple-negative breast cancer

## Abstract

**Simple Summary:**

Previous studies have hypothesized that patients with a more unfavorable oncological prognosis may present with a diminished ovarian reserve, potentially leading to poorer reproductive outcomes even prior to the initiation of gonadotoxic therapies. In the present prospective study involving 152 patients, we examined the relationship between the immunohistochemical profile of breast tumors (hormone receptors, HER2 receptor, and proliferative markers), ovarian reserve indices (AMH, antral follicular count), and the response to fertility preservation procedures (number of mature oocytes retrieved). Our analysis did not show significant differences between patients with triple-negative breast cancer and those with hormone receptor-positive disease.

**Abstract:**

Background: Breast cancer (BC) in reproductive-age women raises concerns about fertility preservation, particularly as systemic therapies may compromise ovarian function. Evidence on whether tumor immunohistotype influences ovarian reserve and fertility preservation outcomes remains limited. Methods: We conducted a prospective cohort study of BC patients referred for fertility preservation counseling between November 2020 and May 2025. Ovarian reserve was assessed using anti-Müllerian hormone (AMH) levels and antral follicle count (AFC). Controlled ovarian stimulation (COS) and oocyte cryopreservation were performed according to standardized protocols. Patients were stratified into triple-negative BC (TNBC) and hormone receptor-positive (HR+) and/or HER2+ groups. The primary endpoint was ovarian reserve differences by subtype; the secondary endpoints were ovarian response and oocyte yield. Results: Of 358 patients, 152 were enrolled, and 139 (91.4%) underwent COS, for a total of 145 cycles. The median age was 33 years, median AMH 5.4 ng/mL, and median AFC 17. No significant differences were observed between the TNBC and HR+/HER2+ groups in AMH, AFC, oocyte yield, or mature oocyte rate. Sub-analysis revealed a significantly lower mature oocyte yield in luminal-B tumors. Conclusions: Ovarian reserve and cryopreservation outcomes appeared preserved in TNBC compared with those in patients with HR+/HER2+ BC at diagnosis. These findings provide reassurance that baseline fertility potential is not compromised by tumor immunohistotype.

## 1. Introduction

Breast cancer is the most common malignancy in women, with a recent significant increase in incidence during the reproductive years. Worldwide, the incidence of breast cancer in women aged 20 to 39 is 144.9 per 100,000 women [1,2]. Invasive breast cancer is a highly heterogeneous disease, and its prognosis primarily depends on both histopathological and molecular features. The most common prognostic factors include the expression of hormone receptors (i.e., estrogen, progesterone, androgen receptors), HER2 status, primary tumor size, and grade, as well as the presence of vascular invasion and lymph node metastases [3,4]. Recent medical developments in diagnosis and treatment have enabled personalized disease management, which has improved patient life expectancy. For young women diagnosed with breast cancer, achieving parenthood can be a critical concern. Indeed, following systemic antiblastic treatments, female fertility is frequently compromised, often resulting in either transient or permanent iatrogenic amenorrhea, as well as early ovarian insufficiency. Some studies have shown that reproductive-age women with malignancies have lower serum levels of anti-Müllerian hormone (AMH) than healthy controls, even prior to starting chemotherapy [5,6,7]. In addition to the cytotoxic damage caused by therapy, recent studies have indicated that women of reproductive age with malignant tumors exhibit lower basal AMH levels than controls, even before the initiation of chemotherapy. Furthermore, during controlled ovarian stimulation (COS) for fertility preservation, women with advanced-stage disease typically require higher doses of gonadotropins than those with low-stage tumors, which may reflect a lower ovarian reserve [3,8]. Larger studies have also reported that circulating AMH concentrations may vary according to genetic mutations, with lower AMH levels observed in BRCA1/2 mutation carriers than those in age-matched controls [9,10].

The aim of this study is to prospectively investigate the influence of the immunohistotype (hormone receptors, HER2 receptor, and proliferative markers) on ovarian reserve and outcome of fertility preservation procedures.

## 2. Materials and Methods

### 2.1. Study Design and Population

This prospective observational cohort study involved patients with breast cancer referred for fertility preservation counseling at a third-level, university-affiliated fertility center from November 2020 to May 2025. Women of childbearing age awaiting breast surgery at the Breast Unit of our institute were invited to participate in the study. As part of the diagnostic work-up at the Humanitas Cancer Center, the patients underwent a gynecological evaluation: during this visit, fertility preservation counseling was offered, and assessment of ovarian reserve [including pelvic ultrasound with antral follicle count (AFC) and serum levels of FSH and AMH] was prescribed. This assessment was conducted independently of the menstrual cycle phase. Patients who accepted the counseling were scheduled for a COS and oocyte cryopreservation. Ovarian reserve indices were measured before the initiation of COS, and all patients were evaluated and underwent fertility preservation procedures prior to the start of any oncologic treatment. Figure 1 shows the flowchart of the procedures.

The following exclusion criteria were applied: previous ovarian surgery, prior infertility treatments, and previous chemotherapy for other malignancies.

COS was initiated, when possible, in the early follicular phase of the ovarian cycle (days 2–3) or randomly [11]. In the latter case, patients were administered estrogen–progestin pills for three days or gonadotropin-releasing hormone (GnRH) antagonists for three days if they had estrogen receptor-positive breast cancer. Patients underwent a GnRH-antagonist cycle with recombinant FSH (rFSH) or human menopausal gonadotropin (hMG), with dosing tailored to age, weight, and ovarian reserve (OR). When the leading follicle reached ≥12 mm, 0.25 mg of Ganirelix or Cetrorelix was added. Estrogen receptor-positive breast cancer patients were also given 5 mg of Letrozole daily from the first day of stimulation until seven days after oocyte retrieval. Follicular growth was monitored daily or every other day using transvaginal ultrasound and measuring 17β-estradiol and progesterone. Once at least two follicles reached 16 mm in diameter, 0.25 mg of recombinant hCG or 0.4 mg of Triptorelin was administered to trigger final oocyte maturation. Dual stimulation, or “duostim,” was proposed for “poor responders” who had time for an additional procedure, based on the number of oocytes retrieved in the first cycle and the patient’s age [11]. Thirty-six hours after final oocyte maturation was triggered, oocyte retrieval was performed under deep sedation in an operating room, using a single- or double-lumen aspiration needle, as previously described [12]. Oocyte nuclear maturity was evaluated following enzymatic denudation, and metaphase II oocytes (MIIs) deemed morphologically suitable were cryopreserved using an open vitrification system with media provided by Irvine Scientific [13]. All costs of the procedures were covered by the Italian National Healthcare System, including the gonadotropins.

The immunohistochemical parameters considered were as follows: estrogen receptors (ER), progesterone receptors (PR), membrane reactivity for HER2/neu, and expression of the nuclear antigen Ki-67. ER and PR expression were assessed using rabbit monoclonal primary antibodies specific to the C-terminal region of ERα (CONFIRM™ anti-ER (SP1) Roche, Switzerland)and PR (CONFIRM™ anti-PR (1E2) Roche, Switzerland), respectively. The same immunohistochemical technique was used to evaluate the proliferation index by assessing Ki-67 expression (CONFIRM™ ANTI-Ki-67 30-9, Roche, Switzerland). Detection was performed using a secondary anti-immunoglobulin antibody-conjugated visualization system. Results were reported as the percentage of positively stained tumor cell nuclei. HER2/neu membrane immunoreactivity was evaluated by immunohistochemistry using anti-HER2 antibodies (c-erbB2, VENTANA Roche, 4B5, Switzerland). In cases with equivocal immunohistochemical results (IHC 2+), HER2 gene amplification status was determined by fluorescence in situ hybridization (FISH).

The primary endpoint is to define the difference in term of ovarian reserve, assessed with AMH dosage and ultrasound AFC, in patients with breast cancer based on their immunohistochemical subtype: patients with triple-negative breast cancer (TNBC) vs. patients with estrogen or progesterone receptor positivity and/or HER-2. The secondary objective is to collect cumulative data on ovarian response and oocyte retrieval after COS in patients who decide to undergo fertility preservation. The comparison between TNBC and HR+/HER2+ groups was pre-specified, as it represents the primary analysis based on our study hypothesis. The subgroup analyses among luminal A, luminal B, and HER2+ patients, were exploratory.

### 2.2. Data Collection

Data regarding patients’ oncological history and oocyte cryopreservation cycles were retrieved from the fertility center’s web-based registry (Art-it). The dataset is regularly updated every three months, including thaw cycles and patient deaths. Pathology reports from biopsies performed for cancer diagnosis were obtained from the internal web-based program (W—Hospital). Patients’ data are safeguarded by advanced threat prevention, enterprise-class encryption, and authentication for any user with the periodical need of password renewal.

### 2.3. Ethical Approval and Data Protection

Patients included in this study provided written consent for the use of their anonymized medical records for research and follow-up, ensuring the confidentiality of their medical data. The procedures of the study adhered to the Declaration of Helsinki ethical principles for medical research involving human subjects. The study was approved by the internal ethics committee with the number ICH2524.

### 2.4. Statistical Analysis

All statistical analyses were performed using STATA version 18.0 (StataCorp, College Station, TX, USA, 2023). Categorical variables were summarized as absolute numbers and percentages and continuous ones as mean and standard deviation. Comparisons between groups were conducted using the χ^2^ test for categorical variables and the Wilcoxon rank-sum test for continuous variables. For patients who underwent double stimulation, each retrieval cycle was evaluated independently. A two-sided *p*-value < 0.05 was considered statistically significant.

## 3. Results

Between November 2020 and May 2025, a total of 358 patients were invited to undergo fertility preservation counseling for breast cancer diagnosis. Among these, 186 patients met the inclusion criteria (51.9%): specifically, having undergone surgical intervention, histopathological examination, and hormonal analysis at our institution. Of the eligible patients, 150 agreed to participate in the study (80.6%). Among the enrolled patients 137 (91.3%) proceeded with COS and oocyte retrieval. Seven poor-responder patients underwent multiple ovarian stimulations (six double and one triple) for a total of 145 stimulation cycles. No surgical complications or ovarian hyperstimulation syndromes (OHSSs) were observed in the study population. Overall, the mean age was 33.7 ± 4.2 years. The mean AMH and the mean AFC were 3.4 ± 2.9 ng/mL and 17.3 ± 10.6 follicles, respectively. After ovarian stimulation a mean of 13.3 ± 8.1 COCs were retrieved. The mean oocyte mature rate (MII/COC) was 64.7%. The cycles’ characteristics are shown in Table 1.

Overall, 25 patients (19.1%) were classified as having TNBC (Group A). The remaining 112 patients were estrogen-, progesterone-, and/or HER-2 receptor-positive (HR+, Group B). The two groups (TNBC and HR+) were comparable in terms of age, with no significant differences observed in ovarian reserve markers (FSH, AMH, AFC), number of COCs retrieved, the number of MII oocytes, and the MII/COC ratio. For patients who underwent multiple stimulations, each retrieval cycle was evaluated independently, all seven of these patients belonged to Group B, HR+. Among the study population, 20% of patients in Group A were carriers of BRCA1 or BRCA2 mutations, compared with 11.8% of those in Group B. A sub-analysis of Group B was then performed, dividing the cohort based on immunohistochemical characteristics into HER2+, luminal A, and luminal B subgroups, as shown in Table 2.

Based on the currently adopted classification system by the pathology department of our institution [14,15], luminal A breast cancer is ER-positive and PR-positive, HER2-negative, and has low levels of the protein Ki-67 (<15%), while luminal B has higher levels of Ki-67 (>15%). This analysis revealed that the sample distribution remained comparable in terms of age, ovarian reserve, and number of retrieved oocytes (Table 2). The only statistically significant finding was a lower rate of MII oocytes retrieved in the luminal B subgroup (51.7 ± 30.3; *p* = 0.049) than those in all other groups included in the analysis. In exploring the potential correlation between cellular proliferation markers (Ki-67) and the MII retrieved oocytes, we plot the Ki-67 expression vs. the number of oocytes (Figure 2), revealing the absence of any linear correlation between the two variables (rho = 0.014). Six patients with luminal A tumors and one with a HER+ tumor underwent double stimulation.

## 4. Discussion

The current study aimed to investigate whether ovarian reserve differs between patients with TNBC and those with HR+ breast cancer. No statistically significant differences were observed in AMH levels or AFC between the two groups. Furthermore, 19.1% of the total sample were classified as having TNBC (Group A), a finding consistent with the existing literature [14]. These results align with previous research that found no significant association between breast cancer subtypes and baseline markers of ovarian reserve prior to the initiation of systemic therapy [16,17]. Recently, Grynberg et al. evaluated [18], in a multicenter retrospective study of 352 patients, the ovarian reserve indices and the response to stimulation of patients with breast cancer, finding no differences in either basal or oocyte recruitment based on the prognostic factors of the tumor (grade, triple-negative status, HER2 expression). Shang-Min Liu et al. [19] evaluated the differences according to the stage of the disease and found no significance in terms of the numbers of mature oocytes or total oocytes retrieved. However, the current evidence remains heterogeneous. Some studies have suggested lower AMH concentrations in patients with more-aggressive disease, including TNBC [20], potentially linked to underlying biological factors such as an increased proliferative index, metabolic alterations, or the presence of BRCA mutations, which have been associated with diminished ovarian reserve [21,22,23]. Differences in study design, sample size, age distribution, and timing of assessment may partially account for these discrepancies. Notably, accumulating evidence suggests that BRCA mutations are associated with a reduced ovarian reserve, independent of a cancer diagnosis. [24]. In our population, BRCA1/2 mutation carriers represented 20% of Group A and 11.8% of Group B; the sample size was insufficient to perform a robust analysis regarding the impact of BRCA status. It is possible that the unequal distribution of BRCA mutations across the two populations (TNBC and HR+) may potentially mask or mimic a subtype-related effect on ovarian reserve, but the limited number of BRCA carriers in our cohort prevents any definitive conclusions regarding its effect. It is important to consider that conducting studies in the field of oncofertility presents significant methodological challenges. In particular, enrolling a sufficiently large and representative sample of young breast cancer patients remains difficult, given the relatively low incidence of the disease in this age group and the urgency of initiating oncological treatment. Furthermore, acquiring reliable data on patients who refuse or are ineligible for fertility preservation is more complex and can lead to selection bias.

Consistent with previously published findings in both IVF settings [25,26] and fertility preservation contexts [27], we observed a positive correlation between ovarian reserve markers and the number of cryopreserved mature oocytes. We observed a recovery rate of mature oocytes of 65.1% and 64.3% in the two groups, respectively. Bayla and colleagues [28] had already reported a lower mature oocyte yield in breast cancer patients, along with a correlation with hormonal receptor status, which we were not able to confirm in our cohort. Tumors with a TNBC profile or with high HER2 expression have also been identified as independent predictors of a suboptimal outcome in in vitro maturation (IVM) techniques [20]. Several mechanisms have been proposed to explain this phenomenon, including systemic inflammatory changes associated with malignancy that could impair granulosa cell function and oocyte maturation [20]. In our cohort, although we observed a trend toward lower mature oocyte numbers in breast cancer patients, we did not confirm a correlation with hormonal receptor status, possibly due to limited sample size. A reduced MII yield was observed in patients with the luminal B subtype as the only statistically significant finding; nonetheless, the difference was minor and could be attributable to chance, considering the sample size limitations. To further investigate this observation, we conducted a correlation analysis between Ki-67 proliferation index and the number of MII retrieved. However, this analysis did not reveal a significant linear trend, suggesting that Ki-67 expression alone may not account for the observed variation in oocyte maturation. To the best of our knowledge, we are the first to conduct a sub-analysis based on the quantitative percentage of Ki-67 expression. This value is commonly used as a binary or categorical marker to differentiate between molecular subtypes or to guide treatment decisions [15]; its role as a continuous variable has been less frequently explored in the context of fertility preservation. However, conducting multiple statistical comparisons may increase the likelihood of type I errors; therefore, the results, particularly from exploratory analyses, should be interpreted with appropriate caution.

A particularly noteworthy finding of our study is the high rate of patients opting for fertility preservation procedures (91.4%). This figure significantly exceeds the rates reported in most published studies, where acceptance among eligible or counseled patients ranges from approximately 30% to 60%. For instance, a French nationwide study reported that, although fertility preservation was discussed with over 60% of young breast cancer patients, only 17% ultimately underwent the procedure [29]. Similarly, in two Italian cohorts 35% and 55%, respectively, of counseled patients accepted fertility preservation after dedicated counseling sessions [30,31]. The exceptionally high adherence rate observed in our cohort may reflect system-level and institutional factors: in our setting, fertility preservation procedures are fully covered by the national health system, and counseling is delivered within a multidisciplinary oncofertility pathway at a tertiary care center. The combination of accurate selection by breast specialists of patients with clear reproductive intentions and the presence of the multidisciplinary counseling team may enhance patient understanding, reduce decisional conflict, and facilitate timely access to cryopreservation techniques.

Nevertheless, ovarian reserve and cryopreservation outcome at diagnosis does not predict long-term fertility outcomes, especially after chemotherapy, which remains a significant gonadotoxic risk regardless of cancer subtype [32]. Therefore, prompt referral to reproductive specialists remains essential in all breast cancer patients of reproductive age. Moreover, these results may help alleviate anxiety in patients who fear that the biological aggressiveness of their tumor may already have affected their reproductive potential.

## 5. Conclusions

Our results show no significant differences in AMH levels, AFC, and oocyte retrieval rates between TNBC and HR+ patients, suggesting that the ovarian reserve is not compromised in TNBC patients compared with their HR+ counterparts at the time of diagnosis. This study is subject to certain limitations, including its single-center design; the relatively small number of patients, particularly in the TNBC group; and possible residual confounding, most notably related to BRCA status. From a clinical perspective, our results are reassuring in suggesting that baseline fertility potential can be preserved in patients with triple-negative breast cancer in the same manner and with the same outcomes as in those with HR+ breast cancer. These findings are of particular interest given the distinct biological and clinical characteristics of TNBC, which is typically associated with a more aggressive disease course, younger age at onset, and limited targeted treatment options.

## Figures and Tables

**Figure 1 cancers-17-03564-f001:**
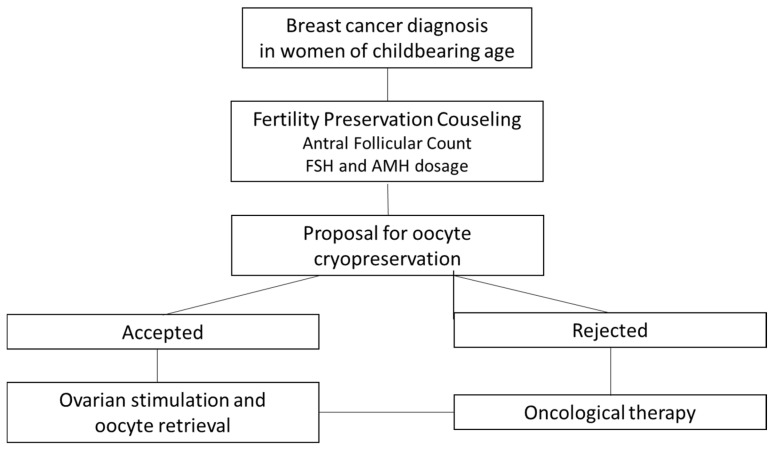
Flowchart of diagnostic and therapeutic procedures.

**Figure 2 cancers-17-03564-f002:**
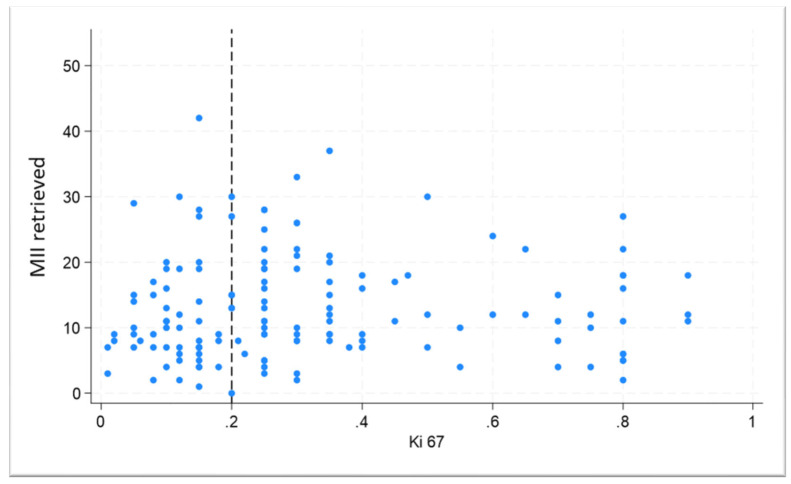
Correlation between Ki-67 expression and the number of metaphase II oocytes retrieved.

**Table 1 cancers-17-03564-t001:** Ovarian reserve parameters and oocyte retrieval outcomes in the triple-negative breast cancer group (TNBC, Group A) vs. the hormone receptor-positive group (HR+, Group B).

Characteristics	Group A—Triple-Negative Breast Cancer	Group B—Hormone Receptor-Positive	*p*
Number of COS cycles	25	120	
Age (years)	32.8 ± 4.6	33.9 ± 4.1	0.235
FSH (mIU/mL)	6.6 ± 3.3	6.0 ± 2.7	0.345
AMH (ng/mL)	2.7 ± 2.0	3.5 ± 3.1	0.256
AFC (*n*)	15.9 ± 9.6	17.6 ± 10.8	0.414
COC retrieved (*n*)	12.8 ± 7.6	13.4 ± 8.2	0.849
MII retrieved (*n*)	8.3 ± 5.2	8.8 ± 6.1	0.827
MII/COC ratio (%)	65.1 ± 19.5	64.5 ± 26.3	0.597
BRCA1/2 mutation carriers (*n*, %)	20.0	11.8	0.246

Footnotes: AFC: antral follicle count; COC: cumulus oocyte complex; MII: metaphase II. For patients who underwent multiple stimulations, each retrieval cycle was evaluated independently. The data are expressed as mean ± standard deviation, if not otherwise specified.

**Table 2 cancers-17-03564-t002:** Ovarian reserve parameters and oocyte retrieval outcomes in triple-negative breast cancer, luminal A, luminal B, and HER2-positive groups.

	Triple-Negative Breast Cancer	Luminal A	Luminal B	HER2-Positive
Number of COS cycles	25	47	40	33
Age (years)	32.8 ± 4.6	35.4 ± 3.8	33.1 ± 3.5	32.9 ± 4.7
FSH (mIU/mL)	6.6 ± 3.3	6.2 ± 3.1	6.0 ± 2.6	5.8 ± 2.4
AMH (ng/mL)	2.7 ± 2.0	2.7 ± 2.6	3.9 ± 3.7	4.4 ± 2.5
AFC (*n*)	15.9 ± 9.6	14.9 ± 9.5	19.1 ± 11.8	19.6 ± 10.8
COC retrieved (*n*)	12.8 ± 7.6	11.7 ± 8.1	13.8 ± 7.7	15.0 ± 9.1
MII retrieved (*n*)	8.3 ± 5.2	7.9 ± 5.8	8.2 ± 6.0	0.7 ± 6.5
MII/COC ratio (%)	65.1 ± 19.5	66.7 ± 25.9	54.9 ± 29.2	72.6 ± 19.7

Footnotes: AFC: antral follicle count; COC: cumulus oocyte complex; MII: metaphase II. For patients who underwent multiple stimulations, each retrieval cycle was evaluated independently. The data are expressed as mean ± standard deviation.

## Data Availability

The dataset generated and analyzed during the current study is available from the corresponding author on reasonable request.

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
