# Peer review of "Does the Immunohistotype of Breast Cancer Influence Ovarian Reserve and Fertility Preservation Outcomes? A Single-Center Prospective Observational Study"

_cancers, 2025, doi:10.3390/cancers17213564_

Round 1
Reviewer 1 Report
Comments and Suggestions for Authors
Very interesting topic with real life consequences. My suggestions for improvement are:
1.The manuscript states a prospective observational case–control design with a defined primary endpoint (AMH/AFC) and secondary endpoints (COS response; MII yield). Please explicitly identify in Methods which analyses were pre-specified (TNBC vs pooled HR+/HER2+) and which were exploratory (luminal A/B/HER2+ sub-groups; Ki-67 correlation). This will help frame the luminal-B finding (p=0.049) as hypothesis-generating.
2.Baseline factors known to influence ovarian reserve/response (age, BRCA status, AMH/AFC at baseline, time-to-COS, letrozole use in ER+ disease, and COS protocol details including random-start and duostim) should be clearly tabulated by subtype and included in the adjustment set. The Statistical Analysis section mentions multivariable logistic regression; please specify covariates, report adjusted effect sizes (ORs) with 95% CIs, and clarify the unit of analysis (per patient vs per cycle) for adjusted models, given that six “poor responders” underwent duostim and cycles were analyzed independently. This transparency is important for internal validity and independence of observations.
3.With multiple endpoints (AMH, AFC, COC, MII, MII/COC) and several subtype contrasts, a single p=0.049 for luminal B may reflect chance. If formal correction is not planned, please temper the language (like “nominally lower”) and provide effect sizes with CIs for context (magnitude over significance). A brief sentence acknowledging the risk of false positives from multiple testing would strengthen the conclusions.
4.BRCA status influences AMH and COS outcomes and differs across subtypes in many cohorts. The manuscript notes BRCA1/2 carriers in 20.0% (TNBC) vs 11.8% (HR+/HER2+), and sample size limited formal analysis. Please consider: A. reporting adjusted results including BRCA status (even as sensitivity analysis), or B. adding a clear statement in Limitations that differential BRCA prevalence could mask or mimic subtype effects on ovarian reserve.
5.Because ovarian markers can vary with disease- or treatment-related timelines, please state the time from diagnosis to counseling/COS, whether labs (AMH/AFC) preceded any systemic therapy, and how letrozole co-treatment was handled in ER+ patients during COS. A concise schematic of the workflow (diagnosis → counseling → COS) would improve clinical interpretability.
6.The continuous Ki-67 analysis found no linear correlation with MII. Consider reporting the slope and CI, and (optionally) a non-linear check (like restricted cubic spline) or a tertile/median split to show that lack of association is not a modeling artifact. Alternatively, clearly label this as an exploratory analysis and keep the current figure with a cautious interpretation.
7.The 91.4% COS acceptance is notably higher than many series, likely reflecting the integrated oncofertility pathway and national coverage of costs. Please expand the Discussion to note health-system context and potential selection factors (counseling at a tertiary center), so readers can judge external validity.
8.In Table 1/2, add footnotes clarifying: A. whether AMH/AFC were measured pre-treatment, B. the definition of MII/COC (%), and C. whether duostim cycles are counted twice and how this is handled in statistical testing.
9.Add one sentence highlighting: single-center design, modest TNBC sample (n=29), potential residual confounding (e.g., BRCA; timing; protocol variations), and cycle-level dependence. These do not detract from the main message but set appropriate expectations.
Comments on the Quality of English LanguageMinor English editing needed
Author Response
Very interesting topic with real life consequences. My suggestions for improvement are:
1.The manuscript states a prospective observational case–control design with a defined primary endpoint (AMH/AFC) and secondary endpoints (COS response; MII yield). Please explicitly identify in Methods which analyses were pre-specified (TNBC vs pooled HR+/HER2+) and which were exploratory (luminal A/B/HER2+ sub-groups; Ki-67 correlation). This will help frame the luminal-B finding (p=0.049) as hypothesis-generating.
We agree that distinguishing between pre-specified and exploratory analyses will clarify the study design and the interpretation of our findings. In the revised version, we have explicitly stated it in the Methods section (lines 227 – 229)
2.Baseline factors known to influence ovarian reserve/response (age, BRCA status, AMH/AFC at baseline, time-to-COS, letrozole use in ER+ disease, and COS protocol details including random-start and duostim) should be clearly tabulated by subtype and included in the adjustment set.
We acknowledge that several potentially relevant variables were not included in the analysis. However, given the relatively small sample size, including too many covariates would have risked model overfitting and reduced the reliability of the results. For this reason, we limited the analysis to the main clinical and biological factors most directly related to the study endpoints.
Ovarian reserve indices were measured before the COS, and all patients were evaluated and underwent fertility preservation procedures prior to the start of any oncologic treatment (we have explained it better in the text).
Since consistent evidence from the literature indicates no significant differences in ovarian stimulation outcomes with random-start protocols, double stimulation, addition of letrozole, or different types of gonadotropins, we did not include these additional variables in the analysis. All stimulation cycles were performed using an antagonist protocol, in accordance with current clinical guideline.
The Statistical Analysis section mentions multivariable logistic regression; please specify covariates, report adjusted effect sizes (ORs) with 95% CIs, and clarify the unit of analysis (per patient vs per cycle) for adjusted models, given that six “poor responders” underwent duostim and cycles were analyzed independently. This transparency is important for internal validity and independence of observations.
Sorry, the article doesn’t include any multivariable logistic regression analysis, due to the exploratory nature of the study. we eliminate the sentence.
3.With multiple endpoints (AMH, AFC, COC, MII, MII/COC) and several subtype contrasts, a single p=0.049 for luminal B may reflect chance. If formal correction is not planned, please temper the language (like “nominally lower”) and provide effect sizes with CIs for context (magnitude over significance). A brief sentence acknowledging the risk of false positives from multiple testing would strengthen the conclusions.
We have added a statement in the Discussion section addressing this limitation and we moderated the tone.
4.BRCA status influences AMH and COS outcomes and differs across subtypes in many cohorts. The manuscript notes BRCA1/2 carriers in 20.0% (TNBC) vs 11.8% (HR+/HER2+), and sample size limited formal analysis. Please consider: A. reporting adjusted results including BRCA status (even as sensitivity analysis), or B. adding a clear statement in Limitations that differential BRCA prevalence could mask or mimic subtype effects on ovarian reserve.
We agree with the reviewer that BRCA status could potentially influence the findings. This limitation has been explicitly mentioned in the Discussion, acknowledging that our sample size does not allow for a meaningful analysis of such correlations.
5.Because ovarian markers can vary with disease- or treatment-related timelines, please state the time from diagnosis to counseling/COS, whether labs (AMH/AFC) preceded any systemic therapy, and how letrozole co-treatment was handled in ER+ patients during COS. A concise schematic of the workflow (diagnosis → counseling → COS) would improve clinical interpretability.
We added a flowchat as requiered (figure 1)
6.The continuous Ki-67 analysis found no linear correlation with MII. Consider reporting the slope and CI, and (optionally) a non-linear check (like restricted cubic spline) or a tertile/median split to show that lack of association is not a modeling artifact. Alternatively, clearly label this as an exploratory analysis and keep the current figure with a cautious interpretation.
We added the Rho value in order to explain the lack of correlation
7.The 91.4% COS acceptance is notably higher than many series, likely reflecting the integrated oncofertility pathway and national coverage of costs. Please expand the Discussion to note health-system context and potential selection factors (counseling at a tertiary center), so readers can judge external validity.
We have expanded the Discussion to clarify that the high COS acceptance rate likely reflects our specific health-system context,
8.In Table 1/2, add footnotes clarifying: A. whether AMH/AFC were measured pre-treatment, B. the definition of MII/COC (%), and C. whether duostim cycles are counted twice and how this is handled in statistical testing.
Done
9.Add one sentence highlighting: single-center design, modest TNBC sample (n=29), potential residual confounding (e.g., BRCA; timing; protocol variations), and cycle-level dependence. These do not detract from the main message but set appropriate expectations.
Done
Reviewer 2 Report
Comments and Suggestions for Authors
The manuscript is devoted to clarifying the relationship between the type of breast cancer and ovarian reserve and the result of fertility preservation procedures. The manuscript corresponds to the profile of the journal Cancers and is of interest to specialists in the field of oncology, although many of the conclusions drawn by the authors only confirm the conclusions drawn in previously published works. The review is well-written and logically structured making it easy to read. The manuscript can be recommended for publication in Cancers after minor revision.
Minor comments
1.Please, indicate in the footnotes of the tables, what value is indicated after the ± symbol. Is it the standard deviation?
- Generally, if data are presented in a table, they are not duplicated in the text (lines 167-172).
- For greater clarity, when analyzing the data presented in Table 2, it is worth first pointing out that all the parameters differ insignificantly for the analyzed groups, and only after point out the exception, the MII/COC ratio, which differs significantly.
- Please clearly indicate which groups were subjected to correlative analysis. (lines 190-193). The same in the legend of Figure 1.
- Please, check grammar in “We observed no statistically…” (lines 200-201) and in “In particular growing evidence...” (lines 217-219).
- Please clearly indicate two groups, which you discuss (line 272).
- From the statement, "baseline fertility potential may be preserved in TNBC patients relative to those with HR+ BC" (lines 273-275), it is not clear what exactly you are stating.
- Please think if some values in the tables have too many digits and should be rounded. For example, 65.5 ± 24.2 or 65±24?
Author Response
1.Please, indicate in the footnotes of the tables, what value is indicated after the ± symbol. Is it the standard deviation?
Done
- Generally, if data are presented in a table, they are not duplicated in the text (lines 167-172).
We have sought to simplify the text by eliminating redundant data in the tables.
- For greater clarity, when analyzing the data presented in Table 2, it is worth first pointing out that all the parameters differ insignificantly for the analyzed groups, and only after point out the exception, the MII/COC ratio, which differs significantly.
We have attempted to rephrase the sentence to improve its clarity.
- Please clearly indicate which groups were subjected to correlative analysis. (lines 190-193). The same in the legend of Figure 1.
Done
- Please, check grammar in “We observed no statistically…” (lines 200-201) and in “In particular growing evidence...” (lines 217-219).
Done. We also had the English proofread by a native speaker.
- Please clearly indicate two groups, which you discuss (line 272).
Done
- From the statement, "baseline fertility potential may be preserved in TNBC patients relative to those with HR+ BC" (lines 273-275), it is not clear what exactly you are stating.
We tried to make the sentence clearer
- Please think if some values in the tables have too many digits and should be rounded. For example, 65.5 ± 24.2 or 65±24?
We have decided to retain one decimal place to ensure greater precision.
Round 2
Reviewer 1 Report
Comments and Suggestions for Authors
The authors have addressed all my concerns adequately and I believe the manuscript can be published in the current form
Comments on the Quality of English LanguageEnglish is better.